# Liver blood marker testing in UK primary care: a UK wide cohort study, 2004–2016

Polly Scutt,[1] Lu Ban,[1,2] Tim Card,[1,3] Colin John Crooks,[3,4] Neil Guha,[3,4] Joe West,[1,3] Joanne R Morling  [1,3]

¹Lifespan and Population Health, University of Nottingham, Nottingham, UK
²European Office, Evidera, London, UK
³NIHR Nottingham Biomedical Research Centre, Nottingham University Hospitals NHS Trust and University of Nottingham, Nottingham, UK
⁴Nottingham Digestive Diseases Centre (NDDC), School of Medicine, University of Nottingham, Nottingham, UK

**Correspondence to**
Joanne R Morling;
joanne.morling@nottingham.ac.uk

## ABSTRACT

**Objective** We aimed to determine (1) the temporal trends of liver enzyme testing in UK general practice and (2) how these vary among different subgroups at risk of chronic liver disease (CLD).

**Design** Retrospective cohort study.

**Setting** UK primary care database (Clinical Practice Research Datalink (CPRD)), 2004–2016.

**Participants** Patients aged 18 years or over, registered in the CPRD from 1 January 2004 to 31 December 2016.

**Outcome measures** The frequency of testing recorded within the study period in general practice was calculated for: alanine aminotransferase (ALT); aspartate aminotransferase (AST); gamma glutamyl transferase (GGT); alkaline phosphatase (ALP); bilirubin and platelets. Analyses were conducted in subgroups of patients at high risk of developing liver disease.

**Results** The study cohort included 2 912 066 individuals with median follow-up of 3.2 years. The proportion of patients with at least one measurement for ALT, ALP, bilirubin or platelet test gradually increased over the course of the study period and fell for AST and GGT. By 2016, the proportion of the population receiving one of more tests in that year was: platelet count 28.0%, ALP 26.2%, bilirubin 25.6%, ALT 23.7%, GGT 5.1% and AST 2.2%. Those patients with risk factors for CLD had higher proportions receiving liver marker assessments than those without risk factors.

**Conclusions** The striking finding that AST is now only measured in a fraction of the population has significant implications for routine guidance which frequently expects it. A more nuanced approach where non-invasive markers are targeted towards individuals with risk factors for CLD may be a solution.

## INTRODUCTION

In the UK, liver disease is a significant and growing burden on the National Health Service (NHS) and is the UK's third most common cause of premature mortality[1]; between 2015 and 2017 it caused 26 265 premature deaths in England alone.[2] It is also a significant source of healthcare inequity, with the median age of death differing by 9 years between the most and least deprived quintiles.[3] There has been a 400% increase in liver disease mortality in the population as a whole since 1970 and nearly 500% increases in mortality observed in working age populations over in this period.[4]

Three independent reports since 2014 have highlighted the need for the early detection of chronic liver disease (CLD) including the Chief Medical Officer report (2012),[5] the All-Party Parliamentary Hepatology Group Inquiry[6] and the Lancet commission,[4] in order to allow intervention and change the course of the disease. A number of organisations have now developed guidance advocating the use of non-invasive fibrosis markers in risk stratification.[7–9] Despite this, many existing community diagnostic pathways for detection and onward referral of suspected CLD are based on traditional liver enzyme tests which lack accuracy and result in delays to diagnosis.[10]

The optimal non-invasive fibrosis marker is yet to be determined, however, there are simple algorithms involving easily accessible measures such as aspartate aminotransferase (AST) and platelets that can be conducted in primary care, for example, aspartate to

platelet ratio index,[11] Fibrosis-4 score (FIB-4)[12] and CIRRhosis Using Standard tests (CIRRUS).[13] However, there is little understanding about how liver blood tests are currently used in UK general practice in order to support the implementation of changing practice and policy.

### Strengths and limitations of this study

⇒ Sampling frame: a significant strength is the use of a large national dataset (>15 mill people).
⇒ Data quality: the dataset used (Clinical Practice Research Datalink) known to be representative of the UK population in terms of age, gender and geographical location with robust quality controls.
⇒ Data validity: previously validated code lists were used for the identification of subgroups.
⇒ A key limitation is the lack of information on the indication for testing or the resultant actions which limits interpretation to some degree.
⇒ Since this study only includes people who attend general practice, and some of the individuals at highest risk of chronic liver disease will not be attending. Therefore, we underestimate the proportions potentially identified in systematic testing was employed.

Given the rising prevalence of lifestyle related CLD and growing knowledge of non-invasive fibrosis measures, one could hypothesise that there should have been a shift away from traditional liver blood testing over time (shifting to non-invasive assessment). The aim of this study was to determine (1) the temporal trends of liver blood testing in UK general practice and (2) how these vary among different subgroups at risk of CLD.

## MATERIALS AND METHODS
### Data source
A population-based cohort study was conducted using the Clinical Practice Research Datalink (CPRD). The CPRD contains primary care data on 15.5 million people from 734 practices in the UK and is considered representative of the UK population.[14] Data are anonymised at patient and practice level and contain information on patient demographics, consultations, diagnoses, referrals and prescriptions. Clinical information is entered using READ codes which was a standard clinical terminology system used in the UK. For a subset of English practices (58% of UK CPRD practices), primary care data can be linked with the Hospital Episode Statistics (HES) dataset containing information for all hospital admissions.[15 16] The population for this study consists only of patients from these practices eligible for linkage with the HES dataset. This was a fully anonymised databased study not requiring ethical approval. This use of the data for this study was approved by the Independent Scientific Advisory Committee for CPRD and the Medicines and Healthcare products Regulatory Agency and assigned reference Protocol 19_256.

### Study population
Patients aged 18 years or over, registered in the CPRD from 1 January 2004 to 31 December 2016, and having at least 1 day of registration with a practice eligible for linkage with the HES dataset were eligible for inclusion in the study. Patients with a diagnosis of CLD before the start of their follow-up period were excluded from the population. Patients were followed up starting at the latest of either the day after the date of current registration with their general practitioner (GP) practice, the start of the study period or the date the GP practice was labelled 'up to standard'. Follow-up ended at the earliest of either the date of death, date the patient transferred out of the GP practice, last date of data collection for the GP practice the patient is registered with, the end of the study period or the date of diagnosis with CLD in primary care (see online supplemental table S1).

### Outcomes
The frequency of testing recorded within the study period in general practice for the following liver blood tests was calculated: alanine aminotransferase (ALT); AST; gamma glutamyl transferase (GGT); alkaline phosphatase (ALP); bilirubin and platelet count. These markers were selected as being routinely used in UK primary care for the assessment of liver function. Abnormal results for each test were defined as: ALT result >50 (IU/L); AST result >40 (IU/L); ALP result >130 (IU/L); GGT result >50 (IU/L); bilirubin result >21 (IU/L); platelet result <150 (platelets/mcl).

### Subgroups
Analyses were conducted in the following subgroups of patients at high risk of developing CLD: presence of type 2 diabetes mellitus (T2DM) defined using READ codes (see online supplemental table S1); obesity defined as a body mass index >30 calculated using height and weight measures; use of alcohol defined as excessive use of alcohol using READ codes (see online supplemental table 1) or recorded >14 units per week alcohol consumption. For all subgroups, follow-up for an individual patient started at the date of diagnosis in primary care. Patients who were diagnosed with CLD within their follow-up period had their follow-up shortened to end 3 months before their date of diagnosis with CLD. An analysis of the subgroup of patients not included in any of these high-risk subgroups was also performed.

### Statistical analysis
Characteristics of the population were compared using $\chi^2$ or Student's t-test as appropriate to the data distribution. The frequency of liver blood testing was presented as the proportion of patients with one or more tests out of the total eligible population over the study period. The frequency of abnormal test results was calculated and presented as the proportion of non-missing test results with an abnormal value. The number of tests performed per year on an individual was calculated by dividing the number of tests performed in the individual's follow-up period divided by the total length of their follow-up period. The proportion of patients with an AST test within 6 weeks following an abnormal ALT test result was calculated.

All analyses were conducted overall and stratified by sex, age group (18–29, 30–39, 40–49, 50–59, 60–69, 70–79, 80+ years) and calendar year. Analyses were performed on the whole study population and in the risk subgroups.

Analyses were performed using SAS V.9.4.

### Patient and public involvement
This study involved members of the Nottingham Digestive Diseases Biomedical Research Unit Patient Advisory Group at the following stages: research design and funding application, lay dissemination and discussion of results.

## RESULTS
### Characteristics
The study cohort included 2 912 066 individuals with follow-up during the years 2004–2016 (median follow-up

3.2 years, IQR 1.3–6.9). Of these, the predefined risk factor subgroups contained: 550 185 (19%) with obesity, 384 011 (13%) with excess alcohol use, 120 305 (4%) with T2DM and 2 235 938 (77%) with none of the three risk factors. One thousand four hundred and eighty individuals had all three risk factors.

The most frequently measured blood marker was platelet count, with 49% of patients having at least one platelet count measured during their follow-up period. The least commonly measured was the AST level with only 12% of patients having at least Exeter@123

in their follow-up. For all tests, the prevalence of testing increased with increasing age, with the highest proportion of patients being tested in the 70–80 year age category. Markers were more frequently measured in women and this difference was statistically significant for all markers (p<0.0001). Full details are given in table 1.

Of those participants having tests the median number of tests undertaken each year was 1, however, some individuals had in excess of 100 of the same test per year. Platelet count was most likely to be tested more than once in an individual with the other liver markers being similar (for additional detail see online supplemental table S2).

### Prevalence of marker measurement over time
The proportion of patients in the study population with at least one measurement for ALT, ALP, bilirubin or platelet test gradually increased over the course of the study period (2004–2016) but conversely fell for AST and GGT markers (figure 1 and table 2). By 2016, the proportion of the population receiving one or more of each test in that year was: platelet count 28.0%, ALP 26.2%, bilirubin 25.6%, ALT 23.7%, GGT 5.1% and AST 2.2%.

### Prevalence of abnormal measures
The proportion of all tests being measured as abnormal remained generally static over the study period (figure 2). Of the 3 922 529 (total number) of ALT test, 343 474 (8.8%) had an abnormal value. The first abnormal ALT test for each patient (n=1 60 191) was paired with an AST test measurement within 6 weeks for 13 997 (8.7%). The proportion of measurements with abnormal values for all other markers was also low: AST (7.5%), ALP (7.9%), GGT (24.6%), bilirubin (4.7%), platelets (16.0%) and these proportions remained stable over the study period.

### Risk factor subgroup analyses
The prevalence of liver marker testing over time by the subgroups (no liver risk factors, excess alcohol consumption and/or obesity) showed similar trends to those for the whole population and are shown in online supplemental figure S1 and online supplemental table S3.

People with T2DM had a notably higher prevalence of testing for all markers (eg, in 2016 ALT measured in 68.8% of those with T2DM vs 15.3% and 21.9% of those with alcohol excess and obesity, respectively). However, the rates of decline in measurement of AST and GGT were also faster in those with diabetes than the other

groups; for AST falling from 24.3% in 2004 to 6.5% in 2016 versus 6.5% and 9.8% to 3.0% and 3.4% of those with alcohol excess and obesity, respectively; and for GGT falling from 28.6% in 2004 to 13.1% in 2016 versus 9.7% and 11.8% to 7.3% and 7.7% of those with alcohol excess and obesity, respectively.

People with no risk factors for liver disease had the lowest prevalence of liver marker testing for all markers, however, did still follow the same trends over time—increasing for ALT, ALP, bilirubin and platelets, and falling for AST and GGT.

## DISCUSSION
We found that while the majority of liver blood markers have shown increased rates of use in general practice over the past 10 years there was wide variation by both marker and subgroups of the population. Most notably, the use of AST has fallen to only 2% per annum among all general practice users.

The striking finding that AST is now only measured in a fraction of the population has significant implications for policy and practice. Major international guidelines, including American, European and British[7 17 18] all use non-invasive markers for investigating liver disease at a community level. AST is a critical component of FIB-4 which has been suggested as a first line test; to rule out significant disease. The absence of AST as a routinely collected marker presents a major barrier to the current implementation of pathways that attend to the aforementioned guidelines. This finding is consistent with other publications, where for example, in the assessment of liver fibrosis in individuals with a diagnosis of non-alcoholic fatty liver disease only 11% had the necessary measures to allow the assessment of FIB-4 in the UK (rising to 54% in Catalonia, Spain).[19] Furthermore, we found that <9% of abnormal ALT measurements also had an AST measured within a 6-week window.

The decision to prioritise ALT measurement over AST may have been driven by a push for efficiency savings[20] with ALT being considered more valuable as it is more liver specific. However, AST may be a more sensitive indicator of chronic liver injury[21–23] especially when used as a ratio with ALT. In some regions an AST is automatically added if the ALT measure is abnormal to facilitate the AST/ALT ratio.[24] Over the 12-year period examined nearly 40% of the population had at least one ALT measurement. This far exceeds the proportion of the known population dying prematurely of liver disease (estimated at 26 265 premature deaths in England in 2015–2 01[25]), or the prevalence of recognised hepatic cirrhosis (estimated at 76.3 per 100 000 in 2001).[26] Though the level of CLD in the UK is not known it is unlikely therefore that these tests are all done in those who have it or even are at high risk, and we therefore have to question why they are being performed and the opportunity cost it represents. Existing evidence suggests they are more often measured as part of routine monitoring than for

**Table 1** Characteristics of participants (ever measurements)

| | | Whole population (n) | ALT 1+tests | ALT 1+abnormal | AST 1+tests | AST 1+abnormal | ALP 1+tests | ALP 1+abnormal | GGT 1+tests | GGT 1+abnormal | Bilirubin 1+tests | Bilirubin 1+abnormal | Platelet count 1+tests | Platelet count 1+abnormal |
|---|---|---|---|---|---|---|---|---|---|---|---|---|---|---|
| All | n | 2 912 066 | 1 112 879 | 160 191 | 284 274 | 33 743 | 1 261 596 | 140 433 | 459 754 | 124 475 | 1 246 003 | 99 633 | 1 414 798 | 301 127 |
| | % | | 38.2 | 14.4 | 9.8 | 11.9 | 43.3 | 11.1 | 15.8 | 27.1 | 42.8 | 8.0 | 48.6 | 21.3 |
| Sex | | | | | | | | | | | | | | |
| Male | n | 1 378 945 | 484 471 | 102 962 | 125 241 | 19 852 | 547 936 | 57 766 | 213 668 | 76 266 | 542 549 | 62 838 | 555 508 | 124 432 |
| | % | | 35.1 | 21.3 | 9.1 | 15.9 | 39.7 | 10.5 | 15.5 | 35.7 | 39.3 | 11.6 | 40.3 | 22.4 |
| Female | n | 1 533 121 | 628 408 | 57 229 | 159 033 | 13 891 | 713 660 | 82 667 | 246 086 | 48 209 | 703 454 | 36 795 | 859 290 | 176 695 |
| | % | | 41.0 | 9.1 | 10.4 | 8.7 | 46.5 | 11.6 | 16.1 | 19.6 | 45.9 | 5.2 | 56.0 | 20.6 |
| Age group, years | | | | | | | | | | | | | | |
| 18–29 | n | 1 117 738 | 196 244 | 19 026 | 46 304 | 3311 | 230 084 | 15 495 | 70 498 | 7038 | 225 208 | 15 343 | 328 696 | 65 280 |
| | % | | 17.6 | 9.7 | 4.1 | 7.2 | 20.6 | 6.7 | 6.3 | 10.0 | 20.1 | 6.8 | 29.4 | 19.9 |
| 30–39 | n | 1 000 314 | 246 015 | 35 067 | 61 081 | 6127 | 287 277 | 18 522 | 95 123 | 19 394 | 281 707 | 18 460 | 370 007 | 74 137 |
| | % | | 24.6 | 14.3 | 6.1 | 10.0 | 28.7 | 6.4 | 9.5 | 20.4 | 28.2 | 6.6 | 37.0 | 20.0 |
| 40–49 | n | 718 585 | 266 922 | 42 817 | 66 307 | 7793 | 307 249 | 19 276 | 108 280 | 30 790 | 303 132 | 20 614 | 324 775 | 62 005 |
| | % | | 37.1 | 16.0 | 9.2 | 11.8 | 42.8 | 6.3 | 15.1 | 28.4 | 42.2 | 6.8 | 45.2 | 19.1 |
| 50–59 | n | 476 113 | 221 016 | 36 113 | 54 770 | 72,51 | 251 719 | 24 052 | 91 799 | 30 896 | 248 874 | 16 947 | 247 761 | 47 522 |
| | % | | 46.4 | 16.3 | 11.5 | 13.2 | 52.9 | 9.6 | 19.3 | 33.7 | 52.3 | 6.8 | 52.0 | 19.2 |
| 60–69 | n | 323 139 | 187 210 | 24 880 | 46 118 | 5646 | 210 752 | 25 495 | 77 627 | 25 207 | 208 949 | 16 616 | 200 591 | 39 762 |
| | % | | 57.9 | 13.3 | 14.3 | 12.2 | 65.2 | 12.1 | 24.0 | 32.5 | 64.7 | 8.0 | 62.1 | 19.8 |
| 70–79 | n | 212 186 | 133 592 | 11 776 | 33 557 | 3637 | 150 577 | 23 758 | 54 137 | 16 123 | 149 271 | 13 196 | 145 700 | 31 741 |
| | % | | 63.0 | 8.8 | 15.8 | 10.8 | 71.0 | 15.8 | 25.5 | 29.8 | 70.3 | 8.8 | 68.7 | 21.8 |
| 80+ | n | 182 665 | 106 411 | 6586 | 25 860 | 2436 | 120 156 | 28 719 | 41 325 | 11 671 | 118 880 | 9989 | 123 721 | 28 849 |
| | % | | 58.3 | 6.2 | 14.2 | 9.4 | 65.8 | 23.9 | 22.6 | 28.2 | 65.1 | 8.4 | 67.7 | 23.3 |

ALP, alkaline phosphatase; ALT, alanine aminotransferase; AST, aspartate aminotransferase; GGT, gamma glutamyl transferase.

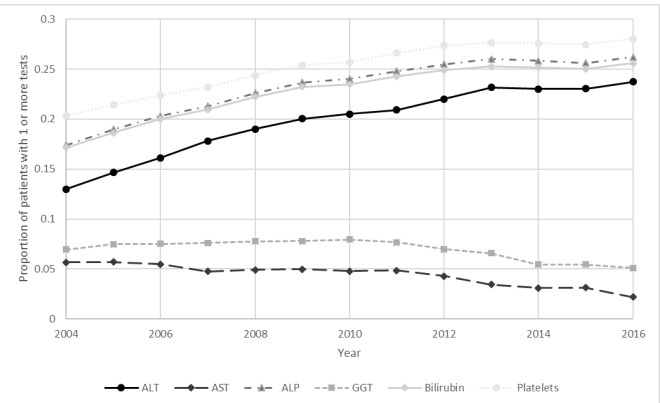

**Figure 1** Prevalence of liver enzyme testing among adults over time. ALP, alkaline phosphatase; ALT, alanine aminotransferase; AST, aspartate aminotransferase; GGT, gamma glutamyl transferase.

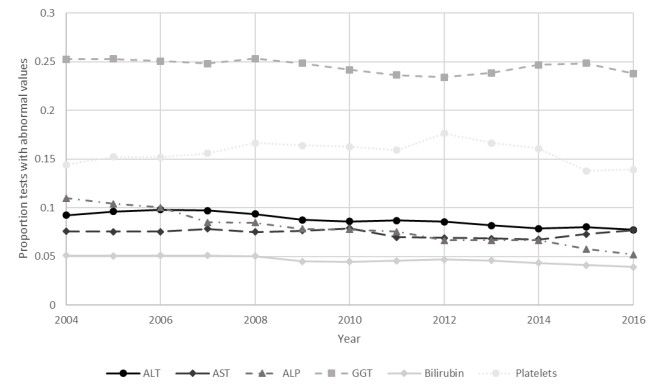

**Figure 2** Prevalence of abnormal values of liver blood tests in adults over time. ALP alkaline phosphatase; ALT alanine aminotransferase; AST aspartate aminotransferase; GGT gamma glutamyl transferase.

CLD identification,[27 28] and that discontinuation of such drugs rarely results.[29] If all these abnormalities were to be followed up (in accordance with existing guidance) there would be significant implications for downstream services. This includes the cost of a full liver screen, liver ultrasound and onward consultation and investigation in secondary care (eg, national tariff for ultrasound scan £75.50, new patient consultant led hepatology outpatient appointment is £208.56.[30] Furthermore, there is growing evidence that in advanced liver disease many individuals have a normal ALT,[10 31] so the growth in use of this marker as a trigger for further assessment may still not identify liver disease.

A more nuanced approach where non-invasive markers are targeted towards individuals with risk factors for CLD may be one solution. From a diagnostic perspective it increases the pre-test probability of having disease and indeed this approach has been shown to be cost effective regardless of choice of biomarker[32 33] and region studied.[34] Within CPRD those patients with risk factors for CLD, as expected, had higher proportions receiving liver markers assessment than those without risk factors. However, this was still very varied by 2016, with 70% of individuals with T2DM having an ALT measure that year, more than double those with obesity and nearly three times those with alcohol excess—with all three groups having similar proportions of abnormal results. While AST testing was more frequent among those with risk factors than in those without it was still very low (<8% in all groups). Therefore, from an implementation perspective

**Table 2** Annual frequency of testing per patient in those with at least one test

|  |  | ALT | AST | GGT | ALP | Bilirubin | Platelet count |
|---|---|---|---|---|---|---|---|
| Median (IQR) maximum number of tests |  | 1 (1–2) 108 | 1 (1–1) 47 | 1 (1–1) 45 | 1 (1–2) 131 | 1 (1–2) 108 | 1 (1–2) 92 |
| 1 | n | 1 914 577 | 436 400 | 691 189 | 2 129 817 | 2 200 429 | 1 972 278 |
|  | % | 74.2 | 75.5 | 75.6 | 70.2 | 74.1 | 60.1 |
| 2 | n | 457 080 | 99 123 | 155 168 | 610 628 | 528 607 | 849 020 |
|  | % | 17.7 | 17.1 | 17.0 | 20.1 | 17.8 | 25.9 |
| 3 | n | 121 616 | 24 862 | 40 197 | 164 364 | 139 762 | 185 136 |
|  | % | 4.7 | 4.3 | 4.4 | 5.4 | 4.7 | 5.6 |
| 4 | n | 39 173 | 8279 | 13 221 | 61 498 | 45 583 | 153 538 |
|  | % | 1.5 | 1.4 | 1.4 | 2.0 | 1.5 | 4.7 |
| 5 | n | 15 569 | 3336 | 4960 | 23 572 | 18 166 | 29 952 |
|  | % | 0.6 | 0.6 | 0.5 | 0.8 | 0.6 | 0.9 |
| 6–10 | n | 22 702 | 4909 | 7084 | 32 002 | 26 262 | 73 577 |
|  | % | 0.9 | 0.8 | 0.8 | 1.1 | 0.9 | 2.2 |
| 11+ | n | 9349 | 1400 | 2610 | 11 234 | 10 248 | 18 286 |
|  | % | 0.4 | 0.2 | 0.3 | 0.4 | 0.3 | 0.6 |

ALP, alkaline phosphatase; ALT, alanine aminotransferase; AST, aspartate aminotransferase; GGT, gamma glutamyl transferase.

it would make sense to focus efforts of obtaining AST and ALT in these groups, appreciating as step change in management is needed.

The strengths of this population approach are driven by the use of a dataset known to be broadly representative of the UK population in terms of age, gender and geographical location with robust quality controls[14] and also the use of validated code lists for subgroup identification.[35] It is therefore reasonable to assume that our findings regarding the level of testing overall and in subgroups are representative of what is happening in the UK. A key limitation is the lack of information on the indication for testing or the resultant actions which clearly limits interpretation to some degree. Additionally, since this study only includes people who attend the GP, some of the individuals at highest risk of CLD will not be attending, the estimates of the proportion of tests which would be abnormal with more systematic testing may be less accurate. A further issue is the lack of information to allow assessment of different liver blood testing systems, for example, which areas 'package' different blood tests together or where abnormal results automatically trigger additional tests.

In conclusion, large numbers of liver blood markers are being measured annually in UK primary care. At present, they are not suitable for risk stratifying high risk populations for CLD as the key element (AST) required to calculate non-invasive fibrosis markers is missing. However, the highest risk groups are receiving regular blood testing (69%% of those with diabetes and 22% of those with obesity) so routine or opportunistic risk stratification could be feasible with limited additional expense to the NHS.

**Contributors** The authors confirm contribution to the paper as follows. JRM, JW, TC contributed to study conception and design. JRM contributed to data collection. JRM, PS, LB, JW, TC, CJC contributed to analysis and interpretation of results. JRM, PS, NG contributed to draft manuscript preparation. All authors reviewed the results and approved the final version of the manuscript. JRM is the guarantor with overall responsibility for the manuscript.

**Funding** This study was funded as part of a Medical Research Council Clinician Scientist award held by JRM (grant number MR/P008348/1).

**Competing interests** None declared.

**Patient and public involvement** Patients and/or the public were involved in the design, or conduct, or reporting, or dissemination plans of this research. Refer to the Methods section for further details.

**Patient consent for publication** Not applicable.

**Ethics approval** Not applicable.

**Provenance and peer review** Not commissioned; externally peer reviewed.

**Data availability statement** Data may be obtained from a third party and are not publicly available. This data was extracting following approval from the Clinical Practice Research Datalink (CPRD, https://www.cprd.com/). Applications for access should be directed directly to CPRD.

**ORCID iD**
Joanne R Morling http://orcid.org/0000-0003-0772-2893

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
