## [Reviewer comments · BMJ Open]

ARTICLE DETAILS

TITLE (PROVISIONAL)	Liver blood marker testing in UK primary care: a UK wide cohort study, 2004-2016
AUTHORS	Scutt, Polly; Ban, Lu; Card, Tim; Crooks, Colin; Guha, Neil; West, Joe; Morling, Joanne

VERSION 1 – REVIEW

REVIEWER	Maja Thiele Odense Universitetshospital, Department for Gastroenterology and Hepatology
REVIEW RETURNED	20-Dec-2021

GENERAL COMMENTS	Review of Liver blood marker testing in UK primary care: a UK wide cohort study, 2004-2016 Overall evaluation: Thank you for the opportunity to review this paper by Scutt P. and co-authors. With this retrospective, population based study, the authors aim to assess the frequency of liver blood samples taken in primary care in the UK, over a 12 year period. Furthermore, they want to investigate the trends in subgroups at risk of liver disease, being obesity, type 2 diabetes and excessive alcohol use. The paper is easy-to-read, and the results are interesting. The authors find that over the 12 year period, 40% of the British population without known chronic liver disease (CLD) have measured at least ALT once. And they find a general tendency towards an increased use of liver blood samples in primary care. However, despite that the blood sample AST is recommended in the large clinical guidelines, to be a part of noninvasive biomarkers, the frequency of which AST is taken in primary care is decreasing over the past decade. There seems to be a controversy is this, and that makes this paper interesting and important. The manuscript's main limitation is the lack of additional data on patient outcomes and testing, to account for actual futile vs relevant testing (for example, good reason to measure ALT in DILI, acute hepatitis and other disease not encompassed by the risk groups defined). There's also a lack of information on diagnostic pathways which would be of added value, for example data on how many patients proceeded to an abdominal US, liver specialist care, elastography; or how many developed cirrhosis due to NAFLD or alcohol related liver disease. MAJOR: Abstract's conclusion: it is not evident for non-hepatologist readers
--

	why the low utilisation of AST has 'significant implications for policy and practice'. One could argue that it's being replaced by ALT in accordance with central regulation. It is not clear why this is not a good trend. Materials and methods: Key definitions are missing. (1) The definition of CLD is not described. Eg. was steatosis enough to have a diagnosis of CLD, and thereby be excluded from the study? (2) the definition of alcohol as a risk factor for CLD is vague. Consider using a less stigmatizing and more specific term than abuse. Abuse is not within ICD-terminology, and it indicates addiction, which is a psychiatric diagnosis, excluding those who exhibit harmful use of alcohol. In the results section you use "excess alcohol use". Materials and methods: The main conclusion is an increase in ALT, bilirubin and platelet testing. Was this increase adjusted for overall population increase per age group, during the same time period? Results: Platelet count is not exclusively a liver blood tests. Authors could also have analysed similar tests, not always associated with liver disease such as INR and albumin? Could you please argue for the choice of tests? (I suspect because they are not part of non-invasive scores APRI, FIB4, CIRRUS, but it is not clear). Results: Data on risk factors could be expanded. For example, you mention that some individuals had more than one risk factor, I am interested in knowing how many? Eg. It is know that having three risk factors, BMI, diabetes and alcohol increases your relative risk more than 10-fold for having advanced liver fibrosis. PMID: 33279778. Results: "By 2016 the proportion of the population receiving one of more tests in that year was", should be ...one or more...? The total sum also only adds up to 111%. This is a surprise to me, since I would expect many tests to come in packages, rarely being measured as sole tests only. For example ALT and ALP. Can the authors explain this finding in more detail? Discussion: There is no discussion of a highly interesting analysis, which unfortunately lacks from the manuscript: if there was an increased detection of CLD in those regions where AST is automatically triggered by abnormal ALT? MINOR: Title page:  - Affiliation no. 2: Clarify Summary:  - P. 3/29 line 11: CLD, write in full length first time Materials and methods:  - P. 6/29 line 20, "...dataset containing information all hospital admissions." Missing adverb in sentence. - P. 6/29 line 24, last sentence in paragraph could be refrased, and what is MHRA? Is "protocol 19_256" from the official numbering system in the MHRA? Otherwise, delete. - P. 7/29 line 27, here you refer to liver enzymes, in the outcomes section you refer to liver blood tests. Please clarify. - P. 7/29 line 52, sentence missing an "at" Results:  - P. 8/29 line 14, consider using an integer for the percentages in the parenthesis.
--	---

	- P. 9/29 line 4-6, the numbers in the parenthesis, seems to differ from the “all (%)” row in table 1? - P. 12/29 line 12, missing an end bracket.
--	--

REVIEWER	William Alazawi Queen Mary University of London
REVIEW RETURNED	31-Jan-2022

GENERAL COMMENTS	This paper addresses an important question in managing liver disease in primary care. Many of the tools proposed to risk stratify liver disease depend upon the ratio of ALT to AST and so understanding the gap between current practice and the ability to implement Fib-4 risk stratification is important. It is worth referencing other work that highlights this same gap in other primary care cohorts. The data contained in the current Table 1 may be better presented as a bar graph and replaced with a table showing the characteristics of the population. An important metric would be the proportion of people in whom a Fib-4 could be calculated (contemporaneous ALT, AST and platelet count) as this would directly highlight the problem at hand.
--

VERSION 1 – AUTHOR RESPONSE

Reviewer 1		
3	The manuscript’s main limitation is the lack of additional data on patient outcomes and testing, to account for actual futile vs relevant testing (for example, good reason to measure ALT in DILI, acute hepatitis and other disease not encompassed by the risk groups defined). There’s also a lack of information on diagnostic pathways which would be of added value, for example data on how many patients proceeded to an abdominal US, liver specialist care, elastography; or how many developed cirrhosis due to NAFLD or alcohol related liver disease.	We thank the reviewer for this comment and agree this is a limitation of the piece. Unfortunately, additional data regarding the purpose and consequences of testing is not available in this dataset. We have noted this in the original manuscript discussion in paragraph 5.
4	Abstract’s conclusion: it is not evident for non-hepatologist readers why the low utilisation of AST has ‘significant implications for policy and practice’. One could argue that it’s being replaced by ALT in accordance with central regulation. It is not clear why this is not a good trend.	We have revised this sentence to provide clarity to the non-hepatologist that current guidance now often requires the measurement of AST. “The striking finding that AST is now only measured in a fraction of the population has significant implications for routine guidance which frequently expects it.”
5	Materials and methods: Key definitions are missing. (1) The definition of CLD is not described. Eg. was steatosis enough to have a diagnosis of CLD, and thereby be excluded from the study? (2) the definition of alcohol as a risk factor for CLD is vague. Consider using a less stigmatizing and more specific term than abuse. Abuse is not within ICD-terminology, and it indicates addiction, which is a psychiatric	(1) We have now directed the reader to the Supplementary Table S1 where the code list for the identification of chronic liver disease was provided. (2) We have clarified this in the text to indicate that alcohol excess was defined as either >14U/wk or through a GP diagnosis of alcohol excess which was recorded using codes

	diagnosis, excluding those who exhibit harmful use of alcohol. In the results section you use “excess alcohol use”.	provided in Supplementary Table S1. We have revised the language as alcohol excess throughout.
6	Materials and methods: The main conclusion is an increase in ALT, bilirubin and platelet testing. Was this increase adjusted for overall population increase per age group, during the same time period?	The main analyses in the paper was not adjusted for the change in age distribution over time. To address this the supplementary joinpoint analyses were age and sex adjusted – and the findings were very similar.
7	Results: Platelet count is not exclusively a liver blood tests. Authors could also have analysed similar tests, not always associated with liver disease such as INR and albumin? Could you please argue for the choice of tests? (I suspect because they are not part of non-invasive scores APRI, FIB4, CIRRUS, but it is not clear).	Choice of blood tests was restricted to those most likely to be undertaken by a GP as part of routine or opportunistic liver assessment. INR/PT are not regularly measured as part of a primary care baseline assessment. We have clarified this choice in the manuscript in the methods paragraph 3. “These markers were selected as being routinely utilised in UK primary care for the assessment of liver function.”
8	Results: Data on risk factors could be expanded. For example, you mention that some individuals had more than one risk factor, I am interested in knowing how many? Eg. It is know that having three risk factors, BMI, diabetes and alcohol increases your relative risk more than 10-fold for having advanced liver fibrosis. PMID: 33279778.	Thank you for his suggestion. We have added to the methods and results (Table S3) the results for individuals with all three risk factors. Those with all three risk factors were most likely to have their liver blood tests measured –the proportion was similar to those with type 2 diabetes and followed the same temporal patterns. It is beyond the scope of this manuscript to investigate the impact of the risk factors on liver fibrosis as we have not measured this. However we agree that it would be the logical next step and we hope to present this in the near future.
9	Results: “By 2016 the proportion of the population receiving one of more tests in that year was”, should be ...one or more...? The total sum also only adds up to 111%. This is a surprise to me, since I would expect many tests to come in packages, rarely being measured as sole tests only. For example ALT and ALP. Can the authors explain this finding in more detail?	This typographical error has been corrected to or . We have clarified the text to indicate that these figures refer to one or more of each type of test – not more than one different test. We agree that the package concept of test ordering seems intuitive. The different ordering structures across the UK are known to be varied and changeable, and

		are not included within the CPRD dataset. In our own local experience ALT, ALP bilirubin are included together and this is illustrated by similar frequencies of testing, with GGT and AST requiring a separate request.
10	Discussion: There is no discussion of a highly interesting analysis, which unfortunately lacks from the manuscript: if there was an increased detection of CLD in those regions where AST is automatically triggered by abnormal ALT?	Unfortunately knowledge of which regions operate an system triggering an AST following and abnormal ALT is unknown. The different ordering structures across the UK are known to be varied and changeable, and are not included within the CPRD dataset. We have revised the discussion to highlight this limitation in paragraph 5. “A further issue is the lack of information to allow assessment of different liver blood testing systems, e.g. which areas ‘package’ different blood tests together or where abnormal results automatically trigger additional tests.” Additionally, analysis of outcomes was beyond the scope of this manuscript.
11	Title page: Affiliation no. 2: Clarify	Updated with full postal address.
12	P. 3/29 line 11: CLD, write in full length first time	Typographical error amended.
13	P. 6/29 line 20, “...dataset containing information all hospital admissions.” Missing adverb in sentence.	Typographical error amended.
14	P. 6/29 line 24, last sentence in paragraph could be refrased, and what is MHRA? Is “protocol 19_256” from the official numbering system in the MHRA? Otherwise, delete.	The Medicines and Healthcare products Regulatory Agency (MHRA) sponsor the CPRD dataset. The protocol ID has been formally assigned by the MHRA. The text of this section has been amended to clarify this.
15	P. 7/29 line 27, here you refer to liver enzymes, in the outcomes section you refer to liver blood tests. Please clarify.	Amended to liver blood tests throughout.
16	P. 7/29 line 52, sentence missing an “at”	Typographical error amended.
17	P. 8/29 line 14, consider using an integer for the percentages in the parenthesis.	Amended as suggested.
18	P. 9/29 line 4-6, the numbers in the parenthesis, seems to differ from the “all (%)” row in table 1?	Typographical error amended.

	parenthesis.	
19	P. 12/29 line 12, missing an end bracket.	Unable to locate missing end bracket.
Reviewer 2		
20	This paper addresses an important question in managing liver disease in primary care. Many of the tools proposed to risk stratify liver disease depend upon the ratio of ALT to AST and so understanding the gap between current practice and the ability to implement Fib-4 risk stratification is important. It is worth referencing other work that highlights this same gap in other primary care cohorts.	Thank you for this insight – we have added additional text to the discussion and cited Alexander et al (BMC Medicine 2018) to the paper.
21	The data contained in the current Table 1 may be better presented as a bar graph and replaced with a table showing the characteristics of the population.	We have reviewed Table 1 and feel that conversion would require multiple bar charts and would detract from seeing all the information together.
22	An important metric would be the proportion of people in whom a Fib-4 could be calculated (contemporaneous ALT, AST and platelet count) as this would directly highlight the problem at hand.	We agree understanding which patients had the available data for a non-invasive fibrosis marker would be the natural next step. This is beyond the scope of this paper as there are multiple fibrosis markers and to include them all would be unwieldy. However, given by 2016 only 2% of the adult population and less than 7% of the high-risk diabetes subgroup had an AST marker performed we can informally conclude that less than this would have the full panel available for computing any fibrosis marker requiring AST.

VERSION 2 – REVIEW

REVIEWER	Maja Thiele Odense Universitetshospital, Department for Gastroenterology and Hepatology
REVIEW RETURNED	17-Mar-2022
GENERAL COMMENTS	The authors have fully clarified my questions, and addressed any concerns. I congratulate them on a very interesting study.